# Joint Effect of Multiple Metals on Hyperuricemia and Their Interaction with Obesity: A Community-Based Cross-Sectional Study in China

**DOI:** 10.3390/nu15030552

**Published:** 2023-01-20

**Authors:** Shan Wu, Huimin Huang, Guiyuan Ji, Lvrong Li, Xiaohui Xing, Ming Dong, Anping Ma, Jiajie Li, Yuan Wei, Dongwei Zhao, Wenjun Ma, Yan Bai, Banghua Wu, Tao Liu, Qingsong Chen

**Affiliations:** 1Guangdong Provincial Engineering Research Center of Public Health Detection and Assessment, School of Public Health, Guangdong Pharmaceutical University, Guangzhou 510310, China; 2Guangdong Provincial Institute of Public Health, Guangdong Provincial Center for Disease Control and Prevention, Guangzhou 511430, China; 3Guangdong Province Hospital for Occupational Disease Prevention and Treatment, Guangzhou 510399, China; 4Department of Public Health and Preventive Medicine, School of Medicine, Jinan University, Guangzhou 510630, China; 5Disease Control and Prevention Institute of Jinan University, Jinan University, Guangzhou 510632, China; 6NMPA Key Laboratory for Technology Research and Evaluation of Pharmacovigilance, Guangdong Pharmaceutical University, 283 Jianghai Avenue, Guangzhou 510300, China

**Keywords:** whole blood metals, hyperuricemia, obesity, cross-sectional study, interaction, BKMR models

## Abstract

Metal exposures have been inconsistently related to the risk of hyperuricemia, and limited research has investigated the interaction between obesity and metals in hyperuricemia. To explore their associations and interaction effects, 3300 participants were enrolled from 11 districts within 1 province in China, and the blood concentrations of 13 metals were measured to assess internal exposure. Multivariable logistic regression, restricted cubic spline (RCS), Bayesian kernel machine regression (BKMR), and interaction analysis were applied in the single- and multi-metal models. In single-metal models, five metals (V, Cr, Mn, Co, and Zn) were positively associated with hyperuricemia in males, but V was negatively associated with hyperuricemia in females. Following the multi-metal logistic regression, the multivariate-adjusted odds ratios (95% confidence intervals) of hyperuricemia were 1.7 (1.18, 2.45) for Cr and 1.76 (1.26, 2.46) for Co in males, and 0.68 (0.47, 0.99) for V in females. For V and Co, RCS models revealed wavy and inverted V-shaped negative associations with female hyperuricemia risk. The BKMR models showed a significant joint effect of multiple metals on hyperuricemia when the concentrations of five metals were at or above their 55th percentile compared to their median values, and V, Cr, Mn, and Co were major contributors to the combined effect. A potential interaction between Cr and obesity and Zn and obesity in increasing the risk of hyperuricemia was observed. Our results suggest that higher levels of Cr and Co may increase male hyperuricemia risk, while higher levels of V may decrease female hyperuricemia risk. Therefore, the management of metal exposure in the environment and diet should be improved to prevent hyperuricemia.

## 1. Introduction

Hyperuricemia has become a severe public health challenge worldwide. It is a potential risk factor for numerous chronic diseases, such as cardiovascular disease, diabetes, kidney disease, and dyslipidemia [1,2]. In China, an increasing prevalence of hyperuricemia has been reported in recent years. It was indicated that there were 13.3% Chinese adults with hyperuricemia in 2014, and this increased to 17.4% in 2019 [3,4]. Moreover, men have a higher risk of hyperuricemia than women, which may be mainly attributed to sex hormones, such as estrogen and testosterone [5,6,7]. In China, the prevalence of hyperuricemia in men was 22.7%, and was significantly higher than that in women (11.0%) [3]. Due to a complex etiology, traditional factors (such as genetics, age, sex, body mass index (BMI), diet, and lifestyle) cannot fully reveal the prevalence of hyperuricemia in the general population [1,5]. Recently, potential associations between environmental exposure to metals and the prevalence of hyperuricemia have attracted considerable attention [8,9]. However, sex-specific heterogeneities in the associations are rarely evaluated.

Metal pollution is a severe environmental issue in China. Accumulating evidence indicates that exposure to metals through water, air, dust, and food may also contribute to the risk of hyperuricemia [10], while the results are still inconclusive. For example, a non-significant association was determined between blood cadmium (Cd) and a risk of hyperuricemia in women from the Survey on Prevalence in East China for Metabolic Diseases and Risk Factors study (SPECT-China) [11], while a cross-sectional study of 4784 general Korean subjects indicated the adverse effects of blood Cd levels only on women’s hyperuricemia risk [8]. Most epidemiological studies have focused on a single or few metals, but people are often exposed to multiple metals in daily life, and the effects of multiple-metal exposures on hyperuricemia risk are largely unknown. A recent study conducted on 1406 Chinese adults indicated that plasma zinc (Zn) was not related to the risk of hyperuricemia, whereas Zn and arsenic (As) were positively associated with hyperuricemia risk or uric acid (UA) levels in the multi-metal logistic regression analysis [9]. The toxicity of a single metal might be enhanced or weakened through inter-metal interactions. The impacts of multiple metals should therefore be considered.

As a widely known risk factor for higher UA and probability of hyperuricemia, a high BMI is reported to be associated with low levels of lead (Pb) and other heavy metals [12]. The evidence at present suggests that obesity is positively correlated with UA levels and hyperuricemia risk [13]. Increased exogenous protein consumption and endogenous UA production in obese persons are additional factors that lead to hyperuricemia [12]. Exposure to heavy metals, such as chromium (Cr), may affect lipid metabolism [14,15]. Thus, the effect of the interaction of metals and BMI on hyperuricemia is worthy of further investigations.

In the present study, we explore the associations of 13 metals in whole blood with the risk of hyperuricemia, as well as potentially sex-dependent heterogeneities and interactions among metal mixtures through a community-based cross-sectional study in China. We measured whole blood samples for the levels of environmental exposure to 13 metals, since blood metals can more truly reflect the internal exposure dose of metals. Multivariate logistic regression models, restricted cubic spline (RCS) models, Bayesian kernel machine regression (BKMR), and interaction analysis were applied to assess the joint effect of multiple metals on whole blood and hyperuricemia risk and their interaction with obesity.

## 2. Methods

### 2.1. Study Settings and Participant Selection

This study used data obtained from the Guangdong Provincial Residents’ Chronic Disease and Nutrition Surveillance Survey (2015). Detailed information on the sampling and survey methods was previously provided [16]. Briefly, the project was conducted in Guangdong Province across 11 selected districts with various degrees of economic development using multistage sampling. Three neighborhoods or townships were randomly selected in each district, and two communities or villages were also selected in each neighborhood or township using probability proportional to size sampling. Of the 3063 eligible individuals, 3029 participants were included in the final analysis after all individuals who did not provide blood samples were excluded. All participants were recruited from general communities, and they were not exposed to metal pollutions or occupational factors. The protocol was approved by the institutional review board of the Chinese Center for Disease Control and Prevention (No. 201519-B), and written informed consent was obtained from all subjects.

### 2.2. Data Collection and Biological Sample Testing

Data collection was described in detail in a previous study [16]. Briefly, this population-based survey covered a health interview and physical examination. Information on demographic and socioeconomic status, lifestyle, medication, diseases, and biochemical profiles were collected. Self-reported diseases were confirmed by medical histories from district-level or higher hospitals.

The anthropometric features included systolic blood pressure (SBP), diastolic blood pressure (DBP), weight, height, and waist measurements. Fasting blood samples of participants were collected and stored at −80 °C for subsequent detection. UA, total cholesterol (TC), triglycerides (TGs), low-density lipoprotein cholesterol (LDL), and fasting blood glucose (FBG) were measured on a Hitachi 7600 automatic biochemical analyzer (Hitachi, Tokyo, Japan).

### 2.3. Outcome and Covariation Evaluation

Hyperuricemia was defined as the following criteria: UA > 420 µmol/L (males) or UA > 360 µmol/L (females), a self-reported history of disease diagnosis, or taking medication [17]. Diabetes was defined as FBG ≥ 7.0 mmol/L, self-reported disease history, or taking diabetes drugs [18]. Hypertension was defined as BP ≥ 140/90 mmHg for three measurements, self-reported history of disease diagnosis, or taking anti-hypertensive drugs [19]. Hyperlipidemia was defined as TC ≥ 6.2 mmol/L, TG ≥ 2.3 mmol/L, low-density lipoprotein cholesterol ≥ 4.1 mmol/L, self-reported history of disease diagnosis, or taking lipid-lowering drugs [20]. Central obesity was defined as waist > 85 cm (men) or waist > 80 cm (women) [21].

Drinking status was classified based on alcohol consumption based on drinking status at present (any alcohol consumption in the past 30 days) and not drinking. Smoking status was classified as “nonsmokers”, “former smokers” (no tobacco use in the past 30 days), or “current smokers” (any tobacco use in the past 30 days). Sitting time was classified as “<4 h/day”, “4 to <6 h/day”, “6 to <8 h/day”, or “≥8 h/day”. Physical activity was defined as performing moderate- or high-intensity physical activity at least once per week.

### 2.4. Whole Blood Metal Concentration Detection

The concentrations of 13 metals, including Cr, Cd, Pb, Zn, As, vanadium (V), manganese (Mn), cobalt (Co), nickel (Ni), copper (Cu), cesium (Se), molybdenum (Mo), and thallium (Tl) in whole blood were detected with ICP-MS (Agilent 7500ce, Agilent Technologies, Santa Clara, CA, USA). Before analysis, frozen whole blood samples were thawed at room temperature (22 ± 5 °C). After unfreezing and shaking in oscillations, 0.25 mL whole blood samples were placed into 5.0 mL tubes, then 4.75 mL of diluent was added (including 2.5 mL HNO_3_ (guaranteed reagent, Thermo Fisher Scientific, Waltham, MA, USA) and 0.5 mL Triton^TM^X-100 (analytical reagent, Sigma, St. Louis, MO, USA)) and diluted to 1000 mL with ultrapure water for the final analysis. Trace Elements Blood L-1~2 (Seronorm, Sero, Oslo, Norway) were used as quality control samples, and quality control was run in each batch (25 samples); the relative standard deviation (RSD) was less than 10% (Appendix A). Moreover, three blank samples (1% HNO_3_) were used to control the potential contaminations and set into each batch. Meanwhile, the spiked recovery method was used by measuring each sample twice, and the recovery rates of all metals were from 90.0% to 110.0%. Once the measured values were suggested to be contaminated or differed from the certified values, the instrument was recalibrated and the previous batch of samples was reanalyzed. The data were multiplied by the dilution factor to obtain the final concentrations. Concentrations below the limits of detection (LODs) were substituted with LOD/$\sqrt{2}$. The LODs for the 13 metals ranged from 0.2 (Co, Cd, Tl) to 20.0 (Zn, Pb) μg/L. Since the proportion of participants whose whole blood Tl and Mo with concentrations lower than LOD was 88.7% and 48.04%, Tl and Mo were not analyzed further. For other metals, more than 70.91% of the samples had values above the LOD (Appendix A).

### 2.5. Statistical Analyses

Since the prevalence of hyperuricemia was significantly different between men and women, the general characteristics of the participants were compared by sex and hyperuricemia state [22]. Normally distributed data are presented as mean ± standard deviation (SD), and variables with skewed distributions are reported as medians with interquartile ranges (IQRs). Categorical variables are presented as percentages. The levels of various metals were transformed into natural logarithms (lns) to approximate normal distributions. Student’s t-, chi-squared, and non-parametric Mann–Whitney U tests were used, depending on data distribution. Pearson correlation tests were applied to estimate the relationships between whole blood metals.

Multivariate logistic regression models were applied to assess the associations of whole blood metal levels with the risk of hyperuricemia. The lowest quartiles (Q1) of metal levels were considered as the reference groups. In the single-metal models, confounding factors, including age, BMI, TC, TG, education level, residence area, smoking status, alcohol use, physical activity, sitting time, diabetes mellitus, and hypertension, were adjusted. Consistently, multi-metal logistic regression models were constructed to assess the associations of multiple-metal concentrations with the risk of hyperuricemia. A backward elimination procedure was conducted, and variables with adjusted *p* < 0.10 were retained in the final model. The trend test was conducted by taking the median of each metal quartile as a continuous variable in the model, and the false discovery rate (FDR) correction was used to adjust for multiple tests.

Restricted cubic spline (RCS) regression models were used to further estimate the dose–response relationship of whole blood metal concentrations with the risk of hyperuricemia, with the percentiles of metal concentrations on logarithmic transformations as the knots. Adjusted factors were consistent with the single-metal model. The *p*-value for non-linearity < 0.05 indicated a non-linear association; otherwise, a linear association was suggested.

Given the non-linearity and interactions of multiple-metal exposure on hyperuricemia, a BKMR analysis was conducted to estimate the joint effect of multiple metals [23]. Metals with significant *p*-values in single-metal models were included in the BKMR analysis. Adjusted factors were consistent with the multi-metal models. First, the cumulative effects of multiple metals on hyperuricemia were assessed by comparing a particular percentile of multiple metals against their median values. Then, exposure–response curves were used to estimate the association of a single-metal level with the risk of hyperuricemia, while maintaining the other remaining metals at median levels. Thirdly, multiple metals were grouped based on correlation coefficients, and the group posterior inclusion probability (group PIPs) and conditional posterior inclusion probability (con PIPs) were calculated to identify the key metals for the risk of hyperuricemia, and the threshold for PIP was 0.5 [24]. Finally, the odds ratios (ORs) of hyperuricemia risk for each metal were estimated when the other metals achieved a particular percentile value (including 25th, 50th, or 75th percentiles). Sensitivity analyses for BKMR were performed by changing the smoothing parameters (b = 50 and b = 200) [25].

Subgroup analysis was performed by including the whole blood metal concentrations (quartiles) and covariates in logistic regression models, and the interaction analysis was tested using stratification variates. The stratification variates included BMI (<28, ≥28 kg/m^2^), waist height ratio (WHt R; <0.6, ≥0.6), central obesity (yes, no), education level (below high school, high school, and beyond high school), alcohol use (drinker, non-drinker), smoking status (current smokers, former smokers, nonsmokers), diabetes mellitus (yes, no), and hypertension (yes, no). The effects of the potential interactions between multiple metals and between multiple metals and other variates on hyperuricemia risk were assessed on an additive scale with the relative excess risk index (RERI), attributable proportion (AP), synergy index (S), and a multiplicative SPSS program with ORs [26].

Statistical analyses were performed using SPSS version 22.0 (IBM Corporation, Armonk, NY, USA) and R Studio (v.4.1.0). Two-tailed *p* < 0.05 indicated significant differences.

## 3. Results

### 3.1. Participant Characteristics and Whole Blood Metal Levels

As shown in Table 1 and Appendix A, the prevalence rates of males and females were 31.6% and 19.4%, respectively. The median levels of UA in the hyperuricemia group (total: 448.3 μmol/L, males: 476 μmol/L, females: 405 μmol/L) were significantly higher than those in the non-hyperuricemia group (total: 301.2 μmol/L, males: 341.8 μmol/L, females: 275.1 μmol/L; *p* < 0.001). The participants with hyperuricemia had higher BMI, WHt R, TC, TG, and prevalence rates of hypertension and central obesity than those without hyperuricemia (all *p* < 0.001). The female participants with hyperuricemia were more likely to be older in age, with a higher prevalence of diabetes mellitus compared to those without hyperuricemia. The male participants with hyperuricemia tended to be highly educated, drink more alcohol, and be nonsmokers (all *p* < 0.05). The whole blood levels of V, Cr, Mn, Co, Cu, and Zn were significantly higher in the male hyperuricemia group compared to the male non-hyperuricemia group (all *p* < 0.05). The whole blood levels of As, Se, and Pb were significantly higher, but Co was moderately lower in the female hyperuricemia group compared to the female non-hyperuricemia group (all *p* < 0.05). No differences in the whole blood levels of Ni, Mo, Cd, and Tl were observed between the hyperuricemia and non-hyperuricemia groups for either sex (all *p* > 0.05).

### 3.2. Associations of Whole Blood Metal Quartiles with Hyperuricemia Risk

Pearson correlation coefficients (*r*) of metals ranged from −0.904 to 0.539 (Appendix A). In males, the multivariate-adjusted ORs (95% CIs) of hyperuricemia were 1.58 (1.1, 2.26) for V, 1.80 (1.27, 2.54) for Cr, 1.68(1.18, 2.39) for Mn, 1.7 (1.23, 2.34) for Co, and 1.55 (1.09, 2.19) for Zn, comparing the highest quartiles with the lowest quartiles of metals (all *p*-trend < 0.05) in the single-metal models (Figure 1). Following FDR adjustments, the *p*-trends for whole blood V, Cr, Mn, and Co were still statistically significant (all *p*-_FDR_ < 0.05). In females, the adjusted OR (95% CI) of hyperuricemia was 0.68 (0.47, 0.99) for V.

In the multi-metal models (Table 2), the OR (95% CI) of male hyperuricemia was 1.70 (1.18, 2.45) for Cr and 1.76 (1.26, 2.46) for Co, while the OR (95% CI) of female hyperuricemia was 0.68 (0.47, 0.99) for V. Moreover, positive trends with increased ORs for male hyperuricemia cases among quartiles of Cr and Co were observed (all *p*-trend < 0.05).

### 3.3. Dose–Response Relationships of Whole Blood Metal Levels and Hyperuricemia Risk

The RCS models revealed wave- and inverted-V-shaped negative associations of whole blood V and Co levels with the risk of hyperuricemia in female participants (both *p*-values for overall associations < 0.001, *p* for non-linearity = 0.028 and 0.042, respectively) (Figure 2). Negative linear relationships between whole blood Cr levels and hyperuricemia risk were only observed for females (*p* for overall association < 0.001, *p* for non-linearity > 0.05), while positive linear relationships between whole blood V, Cr, Co, and Zn and hyperuricemia risk were only observed for males (all *p* for overall association < 0.001, all *p* for non-linearity > 0.05) (Appendix A).

### 3.4. BKMR Analyses

Since only V was significantly associated with female hyperuricemia in the single model, we further analyzed the cumulative effect of the five metals (Mn, Cr, Co, Zn, and V) in males using the BKMR model. Figure 3A presents the linear relationship between exposure to single metals (Mn, Cr, Co, and V) and hyperuricemia and a “U” association between Zn and hyperuricemia when the other four metal exposures were fixed at the median. The group PIPs values in groups 1 (including V and Cr) and 2 (including Mn and Co) were higher than 0.5, and the con PIPs of Cr (1.00) and Co (0.920) were the highest in their group (Table 3).

Figure 3B presents the significant joint effect of five metals on hyperuricemia risk when all metals are at or above their 55th percentile compared to their median values. Moreover, an increased and positive association between Cr exposure and hyperuricemia (50th vs. 25th) was observed when the other four metals were fixed at different percentiles (25th, 50th, or 75th). Similar results are observed for Co exposure and hyperuricemia (75th vs. 50th) (Figure 3C). Figure 3D presents a potential interaction between Mn and Co, Mn and Cr, Mn and Zn, and Cr and Co for increasing the risk of hyperuricemia. The positive slopes for Co, Cr, or Zn became steep at higher concentrations of blood Mn when the other three metals were at median values. Sensitivity analyses were conducted to assess the robustness of the results of the BKMR models, and effects and trends similar to the main analyses were observed. The results are insensitive to the selection of this smoothing parameter.

### 3.5. Subgroup and Interaction Analyses

In males, significantly positive associations between the highest quartile of Cr and hyperuricemia risk were observed in the non-central obesity group, both subgroups of BMI and WHt R (all *p*-trend < 0.05). Similar relationships between Co and male hyperuricemia risk were observed in the low-BMI group, low-WHt R group, and both central obesity subgroups (Table 4).

To further investigate the effect of the potential interaction between whole blood metal levels and BMI on hyperuricemia risk, whole blood metal levels and BMI were divided into two categories (i.e., Zn: ≥geometric mean [GM] vs. <GM; and BMI: ≥28 kg/m^2^ indicating obesity vs. <28 kg/m^2^ indicating no obesity). On the additive scale, a significantly positive interaction was observed between Cr and obesity with regard to their effect on hyperuricemia (AP = 0.44, 95% CI = 0.08, 0.81; Table 5). Similar results between Zn and obesity were obtained for the additive and multiplicative scale (RERI = 1.67, 95% CI = 0.02, 3.33; AP = 0.56, 95% CI = 0.23, 0.88; Table 5).

No interactions were observed between other metals and obesity (date not shown). There were positive trends of Co and Cu with an increased risk of hyperuricemia among non-smokers, while increasing trends of Cr quartiles for increased risk of hyperuricemia were observed among current smokers (Appendix A). Trends for increased risk of hyperuricemia were observed for high levels of Cr and Co among male drinkers (Appendix A). No interaction was observed between other stratified factors and metals (date not shown). A significantly positive interaction between Cr and Zn was observed with regard to their effect on male hyperuricemia (RERI = 0.49, 95% CI = 0.03, 0.95; AP = 0.34, 95% CI = 0.02, 0.65; Appendix A).

In females, significantly negative associations between the highest quartile of V and hyperuricemia risk were observed in the non-obesity group (BMI < 28 kg/m^2^) (OR = 0.67, 95% CI = 0.45, 1). No interactions were observed between stratified factors and metals (date not shown).

## 4. Discussion

In this study, we explored the relationships of 13 metals in whole blood with the risk of hyperuricemia in the Southern-Chinese general population, and achieved several notable results with various statistical models. The results show that single-metal exposure and mixtures of exposure to Cr, Co, and V affect hyperuricemia risk outcomes. BKMR analyses indicated an interaction between whole blood Cd and Zn, and a positive, joint effect of the mixture of five metals (Cd, Cu, Mg, Mo, and Zn) on hyperuricemia when the concentrations of blood metals were above the 55th percentile.

### 4.1. Vanadium

Vanadium (V) is a metal element that is mainly ingested through the diet and indirect exposure to contaminated environments. Several studies have suggested that a moderate intake of V could regulate lipid metabolism, reduce blood glucose levels, and exert anti-cancer effects [14,27], while excessive V intake can increase potential neurotoxic risks, especially for occupational exposure populations [28]. Few epidemic studies have suggested an association between V exposure and UA level or hyperuricemia risk in the general population, but most were focused on occupational exposure populations. A cross-sectional study of 186 traffic policemen in Wuhan reported that urinary V was negatively correlated with serum UA [29]. In our study, the median whole blood level of V in the total population (0.52 μg/L) was lower than the Western-Chinese population (0.73 μg/L) [30]. The whole blood level of V may be linked to increased male hyperuricemia risk, but this association disappeared in the multiple-metal model. A possible explanation was that whole blood V was significantly associated with whole blood Cr (r = 0.539) and As (r = 0.175). Conversely, the risk of female hyperuricemia in the highest V quartile was 0.68-fold higher, compared to the lowest quartile in both the single- and multiple-metal models. A positive linear association between whole blood V levels and increased hyperuricemia risk was present in males, while a wavy negative association was observed in females. The association between whole blood V concentration and the risk of hyperuricemia may be different between sexes. In vivo studies showed that V could increase UA levels in male rates by preventing ROS generation [31], and UA as a natural antioxidant may increase with higher V concentrations in the whole blood. The study on Wuhan traffic policemen also reported a positive association between plasma V and UA levels among smokers [29]. In our study, the prevalence of smoking in females was much lower than in males, which may have led to the opposite conclusion.

In the subgroup analysis, the negative association between the highest quartile of V concentration and female hyperuricemia risk was more obvious in BMI < 28 kg/m^2^ group using the lowest quartile as a reference, while a positive association was observed in the high V + BMI ≥ 28 group and low V + BMI ≥ 28 group compared to the low V + BMI < 28. Although BMI was higher in females with hyperuricemia than those without, there was no interaction between whole blood V and obesity in females with hyperuricemia. V supplementation in mice with high-fat-diet-induced obesity could inhibit preadipocyte differentiation and adipogenesis through the downregulation of the adipogenic transcription factors PPARγ and C/EBPα and their target genes in 3 T3-L1 adipocytes [14]. Accumulating epidemiological studies also report a significant positive association of UA with obesity in adults [32,33]. Since the average age of females in our study was 51.83 years old, most of them were experiencing menopause where females usually have increased levels of serum UA due to decreased levels of estrogen and progesterone [34]; therefore, the interaction of V with obesity in females with hyperuricemia might have been masked.

### 4.2. Cobalt

Cobalt (Co) is a metal element and a component of vitamin B12 that mainly enters the human body through food consumption and occupational and iatrogenic exposure. Excessive Co can induce neurotoxicity and genetic toxicity, and even increase the risk of cardiovascular diseases [35]. The exposure concentration of Co is associated with hypertension, diabetes, and dyslipidemia [30,36,37]; however, studies conducted on the association between Co exposure and UA are limited. For instance, a cross-sectional study on 1046 participants in a middle-aged and older Chinese population reported that the median plasma level of Co in hyperuricemia was higher than that in the no-hyperuricemia group (0.25 μg/L vs. 0.24 μg/L); however, Co exposure was not significantly associated with hyperuricemia risk [9]. In our study, the median whole blood level of Co in the Southern-Chinese population was 0.28 μg/L, which was much higher than that of the Western-Chinese population (0.14 μg/L) [30]. The whole blood Co concentration of females was also higher than that of males (0.31 μg/L vs. 0.25 μg/L) due to the higher accumulation rate of circulating Co in females [38]. Females with hyperuricemia had lower whole blood Co levels compared to those without hyperuricemia. However, the opposite was true for males. Compared to the lowest quartile value, male subjects with the highest quartile values of Co levels showed 1.70- and 1.76-fold-higher risks of hyperuricemia outcomes in the single- and multi-metal models, respectively. Notably, this association was not observed in females. Males also showed a positive linear association between whole blood Co levels and increased hyperuricemia risk, while an inverted V-shaped negative association was observed for females. A study using 815 pregnant women from Puerto Rico reported that Co exposure was significantly related to an increase in maternal hormones (sex hormone-binding globulin, estriol, and progesterone) [39]. An animal study observed that chronic Co exposure could significantly increase serum testosterone levels [40]. Additionally, UA levels are regulated by sex hormones, estrogen may induce an increase in uric acid excretion [6], and male testosterone levels are positively correlated with SUA. Therefore, Co may influence the risk of hyperuricemia via the regulation of sex-hormone levels.

In the subgroup analysis, the positive association between the highest quartile of Co concentration and hyperuricemia risk was more obvious in participants who were male, had a BMI < 28 kg/m^2^ or WHt R < 0.6, and both subgroups of central obesity. Moreover, no interactions were observed between whole blood Co and obesity in males or females with hyperuricemia. Co may reduce white adipose tissue (WAT) weight and the size of enlarged adipocytes in mice fed a high-fat diet, possibly mediated by AMP-activated protein kinase activation [41]. However, Co did not correct metabolic abnormalities, but did reduce proteinuria and histological kidney injury in an obese, hypertensive, type 2 diabetes rat model [42]. An epidemiologic survey suggested that fingernail Co concentrations in the American female population were negatively correlated with BMI [43]. Thus, the interaction between Co levels and BMI in the context of hyperuricemia remains unclear; further studies are needed.

### 4.3. Chromium

Chromium (Cr) is the most controversial transition metal and occurs primarily in the trivalent (III) or hexavalent (VI) states. Trivalent Cr (III) has been referred to as a glucose-tolerance factor, and is implicated in the regulation of glucose and lipid metabolism [44], while hexavalent Cr (VI) is recognized as a pulmonary carcinogen. Knowledge about its role in living organisms is constantly expanding. A case–control study conducted on 163 patients with type 2 diabetes from the Bolgatanga district of Ghana reported that low Cr levels were associated with high blood pressure, obesity, and lipid dysregulation [15]. Goldfish treated with Cr presented decreased total glutathione in the liver by 34–69% and kidney by 36–49%, indicating that trivalent Cr (III) induces oxidative stress in the liver and kidney [45]. A dose-dependent association between kidney injury molecule 1 across urinary Cr exposure tertiles using multivariate adjusted models was observed (T1: reference, T2: 467 pg/mL, T3: 615 pg/mL; *p*-trend = 0.001) [46]. However, few studies have investigated the association between human blood Cr and UA levels or hyperuricemia risk in the general population. In our study, the median whole blood levels of Cr (male: 5.14 μg/L; female: 4.84 μg/L) were higher than those measured in Wuhan (male: 0.36 μg/L; female: 0.40 μg/L) [47]. Our data also provide some supporting results on the positive association between whole blood Cr and risk of hyperuricemia in the multiple-metals models. There was a positive linear relationship between whole blood Cr concentration and hyperuricemia risk in males, but not in females. In contrast, the cross-sectional study of traffic policemen in Wuhan reported that a 1-IQR increase in urinary Cr was associated with a 27.4 µmol/L (95% CI = −46.1, −8.8; *p* = 0.004) decrease in serum UA levels [29]. Our subgroup analysis showed a synergistic additive interaction between BMI and whole blood Cr levels on male hyperuricemia risk. Compared to the low Cr + BMI < 28 group, the risks of male hyperuricemia in the high Cr + BMI < 28 and high Cr + BMI ≥ 28 groups were increased by 1.541- and 3.68-fold, respectively. Evidence from available randomized controlled trials showed that Cr supplementation induced statistically significant reductions in body weight, but the magnitude of the effect was small with unclear clinical relevance [48]. An in vitro study observed that maternal Cr restriction significantly increased body weight and fat percentage in both male and female offspring by increasing the gene expression of 11 beta-hydroxysteroid dehydrogenase 1 and leptin [49]. Low levels of blood Cr can positively regulate lipid metabolism, while higher levels may induce oxidative stress [45], leading to disorders of fat metabolism and increasing the risk of hyperuricemia.

### 4.4. Other Metals

Zinc (Zn) is a metal element involved in a variety of biological processes as a structural, catalytic, and intracellular signaling component. As an antioxidant and anti-inflammatory agent in humans, Zn may be associated with serum UA, which may function as a pro-oxidant and pro-inflammatory factor under specific circumstances [50,51]. A cross-sectional study of 24,975 American adults aged 20 years or older and a cross-sectional study of 2697 middle-aged and older Chinese males (aged ≥40 years) both reported an inverse association between dietary Zn intake and hyperuricemia risk [52,53]. However, our study presented that the risk of male hyperuricemia in the highest Zn quartile was 1.55-fold higher compared to the lowest quartile using the single-metal model; however, this association disappeared in the multi-metal model, which could be due to the fact that whole blood Zn was significantly correlated with whole blood Cu (r = 0.338) and had a significant positive interaction with the Cr effect on hyperuricemia. A positive linear association between whole blood Zn levels and increased hyperuricemia risk was observed in males. Consistent with our results, a recent study conducted in China indicated that higher plasma levels of Zn might increase hyperuricemia risk and showed positive dose–response relationships [9]. Oral Zn therapy may lead to higher UA due to liver dysfunction and an increased expression of xanthine oxidase in the liver [54,55]. In our subgroup analysis, there was a synergistic interaction between whole blood Zn concentration and BMI on the additive and multiplicative scales. A cross-sectional study conducted on 1896 Korean adults suggested that serum Zn levels increased with higher tertiles of total body fat percentage [56]. Previous studies reported that obese people have lower serum Zn levels, and Zn deficiency may induce the onset of diabetes and metabolic diseases by reducing levels of the Zn transporter ZIP13 [57].

Arsenic (As) is a human carcinogen that is ubiquitous in the environment. Chronic As exposure may be associated with diabetes, kidney disease, and cardiovascular disease [58]. The kidney is an organ that is most sensitive to As [59]. Chronic and excessive As intake may cause oxidative nephropathy by producing reactive oxygen species (ROS), reactive nitrogen species (RNS), and dimethyl As peroxyl radicals, ultimately hindering the excretion of UA [60]. The NHANES study of US adults observed a significant positive association between low urine As levels (≤1 μg/L) and serum UA in males [17]. Consistently, Wang et al. [9] reported a positive dose–response relationship of plasma As with hyperuricemia risk in middle-aged and older Chinese subjects. Our data suggest no association between whole blood As and hyperuricemia. Other studies have implicated As as a risk factor for obesity because it can suppress fat accumulation, promote lipolysis, damage WAT, and ultimately lead to obesity [61]. In our subgroup analysis, the risk of male hyperuricemia in the high As + BMI ≥ 28 group was 2.08-fold higher compared to the low As + BMI < 28 group, but no interactions were observed between whole blood As and obesity in males with hyperuricemia. The interaction between As levels and obesity in hyperuricemia was unclear due to the different As levels and the study design; therefore, further investigations are necessary to validate our results.

Manganese (Mn) is both an essential element and a known toxic heavy metal. Moderate intake could lower blood pressure and reduce the risks of chronic kidney disease and type II diabetes [62,63], but excessive Mn exposure could affect the central nervous system and induce mood disorders, blunted response, and olfactory impairment [64,65]. UA as a neurotransmitter plays a protective role in certain conditions, such as neurodegenerative disease and injury mediated by peroxynitrite [66]. Therefore, it is necessary to measure the UA levels of individuals that are exposed to Mn. An occupational-based study of 65 welding and foundry workers and 29 controls indicated that long-term and low-level occupational Mn exposure may induce lower UA levels in urine [67]. However, a 12-month study conducted on 64 patients with chronic renal failure (CRF) and 62 healthy controls indicated that CRF patients had higher plasma levels of Mn and UA than the controls, and plasma Mn was positively correlated with creatinine, plasma urea, and plasma UA [68]. In our study, the median levels of whole blood Mn (male: 13.33 μg/L, female: 14.61 μg/L, total: 13.9 μg/L) were higher than in the USA (male: 8.68 μg/L, female: 10.09 μg/L) [69]. The risk of male hyperuricemia in the highest Mn quartile was 1.68-fold higher, compared to the lowest quartile in the single-metal model. Aditionally, there was no association between whole blood Mn and hyperuricemia risk in females. This suggests that excessive Mn exposure has a greater influence on men than women. Further general population studies are warranted to clarify the relationship between Mn and UA.

### 4.5. Strengths and Limitations

This study had several advantages. First, it used a relatively large South-population-based Chronic Disease and Nutrition Surveillance Survey dataset (3063 subjects), which ensured adequate statistical power to test the associations between hyperuricemia risk and multiple metals. Secondly, the whole blood levels of multiple metals as an internal exposure biomarker may be more appropriate to reflect long-term nutritional status and metal exposure compared to serum and urine [70,71]. Third, the large sample size allowed us to adjust for many covariates, including demographic characteristics, socioeconomic status, lifestyle behaviors, anthropometric features, and other chronic diseases, which reduced the risk of confounding bias.

Several limitations should also be considered. First, this was a cross-sectional study; therefore, causality between whole blood metal levels and hyperuricemia could not be shown. Secondly, we did not collect food consumption information; however, diet affects the levels of UA and metals in the body to a certain extent. However, multiple statistic models and the sensitivity analysis of the BKMR model further confirmed the robustness of our results. Finally, metal ions with different valences have different biological effects; exploring the complex effects of metal-mixture exposure warrants the detection of the valence states of metals in future studies.

## 5. Conclusions

Our results demonstrate that higher levels of whole blood Cr, Co, Zn Mn, and V may increase hyperuricemia risk in males in the general population of South China. Among these metals, Cr and Co showed positive dose–response relationships with hyperuricemia risk. The positive interaction between blood Cr and Zn concentrations and obesity influenced male hyperuricemia risk. Whole blood V was an independent protective factor for female hyperuricemia risk and showed a nonlinear relationship. These results increase our understanding of the relationship between exposure to multiple metals and hyperuricemia. Additionally, our results have important public health significance and provide scientific evidence for establishing environmental standards or guiding dietary supplementation of trace-metal elements.

## Figures and Tables

**Figure 1 nutrients-15-00552-f001:**
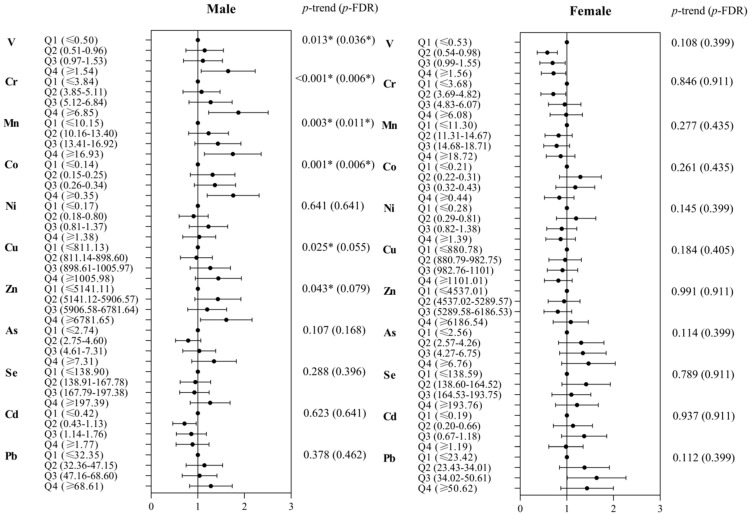
Single-metal logistic regression model for the associations of blood metal quartiles with hyperuricemia. Model is adjusted for age, education level, residence area, smoking status, alcohol use, physical activity, sitting time, BMI, diabetes mellitus, hypertension, TC, and TGs. *p*-trend across quartiles of metals are obtained by including the median of each quartile (natural log-transformed) as a continuous variable in the single-metals model. FDR corrections are performed to adjust for multiple tests. * *p* < 0.05.

**Figure 2 nutrients-15-00552-f002:**
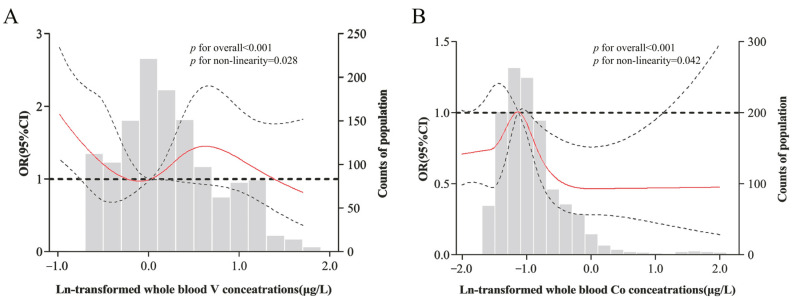
The RCS with five knots for the association between blood metal levels and incident females’ hyperuricemia. The red lines indicate adjusted odds ratios and dotted lines are the 95% confidence intervals based on the restricted cubic spline models for the natural log-transformed concentrations of whole blood V (**A**) and Co (**B**) in females. The reference values were set at the 10th percentiles. Adjusted variables are the same as those listed for the single-metal model.

**Figure 3 nutrients-15-00552-f003:**
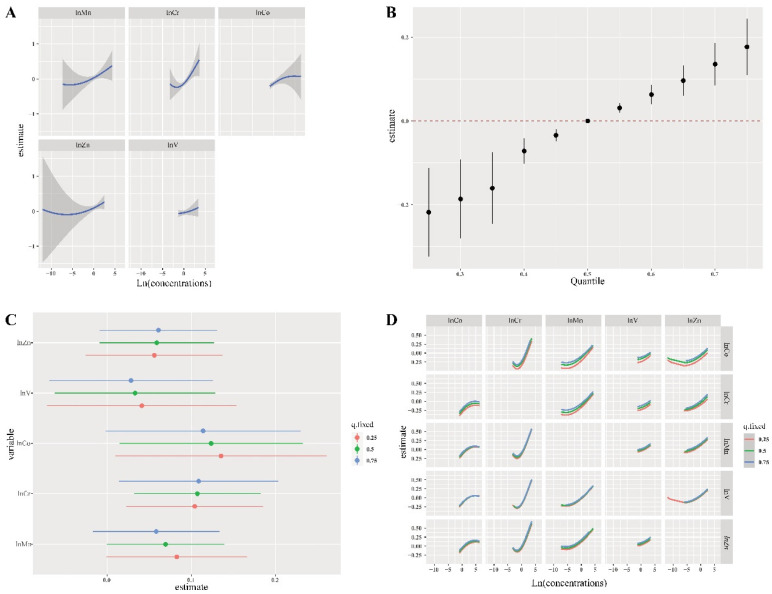
Joint effect of the five metals on male hyperuricemia risk using BKMR. Adjusted variables are the same as those listed for the single-metal model. (**A**) Single-metal exposure–response functions and 95% confidence intervals (shaded areas) for each metal with the other four metals holding at the median. (**B**) Overall effect of the mixture estimates and 95% confidence intervals. (**C**) Single-metal association (estimate and 95% confidence intervals). This plot compares hyperuricemia risk when a single metal is at the 75th vs. 25th percentile, when all the other exposures are fixed at either the 25th, 50th, or 75th percentiles. (**D**) Bivariate exposure response for each of the metals presented on the right longitudinal axis when other metals are presented on the upper coordinate axis holding at different quantiles (25th, 50th, and 75th), and all other metals are held at median values.

**Table 1 nutrients-15-00552-t001:** Baseline characteristics of the study participants.

Variables ^a^	Total		*p*-Value
	No Hyperuricemia (*n* = 2275)	Hyperuricemia (*n* = 754)	
Age (years)	51.97 ± 0.32	54.72 ± 0.53	<0.001 *
Gender (%)			<0.001 *
Male	939 (41.3)	433 (57.4)	
Female	1336 (58.7)	321 (42.6)	
Residence area (%)			0.702
Coastal	1240 (54.5)	417 (55.3)	
Inland	1035 (45.5)	337 (44.7)	
Education level (%)			0.120
Below high school	1694 (74.5)	536 (71.1)	
High school	363 (16.0)	144 (19.1)	
Over high school	218 (9.5)	74 (9.8)	
Alcohol use (%)			0.002 *
Drinker	799 (35.1)	311 (41.2)	
Non-drinker	1476 (64.9)	443 (58.8)	
Smoking status (%)			0.002 *
Current smokers	562 (24.7)	232 (30.8)	
Former smokers	142 (6.2)	52 (6.9)	
Nonsmokers	1571 (69.1)	470 (62.3)	
Sitting time (%)			0.362
<4 h/day	925 (40.7)	284 (37.7)	
4 to <6 h/day	643 (28.3)	212 (28.1)	
6 to <8 h/day	324 (14.2)	122 (16.2)	
≥8 h/day	383 (16.8)	136 (18.0)	
Physical activity (%)			0.023 *
Yes	379 (16.7)	153 (20.3)	
No	1896 (83.3)	601 (79.7)	
WHt R	0.50 (0.46, 0.54)	0.52 (0.49, 0.57)	<0.001 *
BMI (kg/m^2^)	22.66 (20.64, 24.91)	24.24 (22.11, 26.74)	<0.001 *
UA (μmol/L)	301.20 (256.50, 343.20)	448.30 (415.33, 497.75)	<0.001 *
Log_2_UA	8.19 ± 0.01	8.83 ± 0.01	<0.001 *
TC (mmol/L)	4.98 (4.34, 5.61)	5.20 (4.57, 5.97)	<0.001 *
TG (mmol/L)	0.96 (0.68, 1.44)	1.43 (0.98, 2.11)	<0.001 *
Diabetes mellitus (%)			0.005 *
Yes	176 (7.7)	83 (11.0)	
No	2099 (92.3)	671 (89.0)	
Hypertension (%)			<0.001 *
Yes	730 (32.1)	360 (47.7)	
No	1545 (67.9)	394 (52.3)	
Central obesity (%)			<0.001 *
Yes	834 (36.7)	411 (54.5)	
No	1441 (63.3)	343 (45.5)	
Blood metals (μg/L)			
V	0.97 (0.52, 1.51)	1.02 (0.35, 1.62)	0.287
Cr	4.87 (3.71, 6.29)	5.26 (3.86, 7.07)	<0.001 *
Mn	13.87 (10.73, 17.93)	14.14 (10.70, 17.97)	0.466
Co	0.28 (0.14, 0.40)	0.28 (0.14, 0.37)	0.163
Ni	2.24 (1.29, 4.03)	2.21 (1.22, 3.92)	0.483
Cu	943.11 (844.56, 1055.08)	945.20 (848.19, 1044.35)	0.954
Zn	5500.30 (4717.59, 6400.11)	5812.56 (4926.99, 6698.70)	<0.001 *
As	4.38 (2.61, 6.90)	4.87 (2.81, 7.74)	<0.001 *
Se	164.27 (137.04, 193.00)	168.57 (140.39, 201.59)	0.005 *
Mo	1.05 (0.71, 2.05)	1.09 (0.71, 2.27)	0.164
Cd	2.23 (1.33, 4.27)	2.33 (1.23, 4.54)	0.691
Tl	0.14 (0.14, 0.14)	0.14 (0.14, 0.14)	0.535
Pb	38.29 (25.49, 58.64)	42.34 (29.29, 62.24)	<0.001 *

BMI, body mass index; TC, total cholesterol; TGs, triglycerides; UA, uric acid; WHt R, waist height ratio. ^a^ Values are presented as *n* (%), mean ± SD or median (IQR). * *p* < 0.05.

**Table 2 nutrients-15-00552-t002:** Multiple-metal logistic regression model for the associations of blood metal quartiles with hyperuricemia.

Whole Blood Metals ^a^	Q1	Q2	Q3	Q4	*p*-Trend (*p*-FDR) ^b^
Male					
Cr	Ref	0.95 (0.64, 1.41)	1.08 (0.73, 1.60)	1.70 (1.18, 2.45) *	0.001 * (0.005 *)
Co	Ref	1.34 (0.91, 1.96)	1.39 (0.99, 1.95)	1.76 (1.26, 2.46) *	0.002 * (0.005 *)
Cu	Ref	0.81 (0.56, 1.16)	1.14 (0.80, 1.64)	1.30 (0.90, 1.89)	0.054 (0.09)
Zn	Ref	1.47 (1.02, 2.11) *	1.08 (0.74, 1.56)	1.40 (0.97, 2.04)	0.245 (0.245)
As	Ref	0.75 (0.52, 1.09)	0.99 (0.69, 1.41)	1.27 (0.87, 1.85)	0.157 (0.196)
Female					
V	Ref	0.55 (0.38, 0.81) *	0.66 (0.44, 0.98) *	0.68 (0.47, 0.99) *	0.108

^a^ The metals reported in the multiple-metal model are selected using the backward elimination method; adjusted covariates are the same as listed for the single-metal model. ^b^ *p*-trend across quartiles of metals are obtained by including the median of each quartile (natural log-transformed) as a continuous variable in the single-metals model. FDR corrections are performed to adjust for multiple tests. * *p* < 0.05.

**Table 3 nutrients-15-00552-t003:** PIPs for group and conditional inclusions for male hyperuricemia using BKMR model.

Blood Metals (μg/L)	Group	Group PIPs	Con PIPs
V	1	0.932	0.814
Cr	1	0.932	1.000
Mn	2	0.986	0.894
Co	2	0.986	0.920
Zn	3	0.490	0.490

PIPs, posterior inclusion probabilities. Model is adjusted for age, education level, residence area, smoking status, alcohol use, physical activity, sitting time, diabetes mellitus, hypertension, TC, and TGs.

**Table 4 nutrients-15-00552-t004:** Adjusted odds ratios for incident hyperuricemia risk in subgroups stratified by BMI, WHt-R, and central obesity.

	Q1	Q2	Q3	Q4	*p*-Trend ^a^
Male					
Cr (μg/L)	≤3.84	3.85–5.11	5.12–6.84	≥6.85	
Hyperuricemia/total	92/338	93/344	110/343	138/341	
BMI <28 kg/m^2^	Ref	0.96 (0.64, 1.46)	1.08 (0.72, 1.64)	1.66 (1.13, 2.44) *	0.005 *
BMI ≥28 kg/m^2^	Ref	1.81 (0.39, 8.43)	2.11 (0.55, 8.08)	8.38 (1.71, 41.00) *	0.012 *
WHt R <0.6	Ref	0.93 (0.62, 1.39)	1.07 (0.72, 1.59)	1.58 (1.09, 2.30) *	0.008 *
WHt R ≥ 0.6	Ref	1.27 (0.05, 33.69)	6.72 (0.38, 119.71)	165.77 (2.57, 10,707.79) *	0.015 *
Non-central obesity	Ref	1.04 (0.60, 1.81)	1.17 (0.67, 2.03)	1.82 (1.10, 3.02) *	0.012 *
Central obesity	Ref	0.79 (0.44, 1.44)	1.03 (0.59, 1.81)	1.74 (0.99, 3.06)	0.031 *
Co (μg/L)	≤0.14	0.15–0.25	0.26–0.34	≥0.35	
Hyperuricemia/total	140/493	72/218	102/321	119/340	
BMI < 28 kg/m^2^	Ref	1.37 (0.92, 2.04)	1.34 (0.93, 1.92)	1.65 (1.17, 2.35) *	0.006 *
BMI ≥ 28 kg/m^2^	Ref	0.68 (0.14, 3.30)	0.90 (0.23, 3.60)	1.56 (0.39, 6.16)	0.609
WHt R < 0.6	Ref	1.30 (0.88, 1.93)	1.37 (0.97, 1.95)	1.64 (1.16, 2.31) *	0.005 *
WHt R ≥ 0.6	Ref	0.01 (0.01, 4.74)	0.07 (0.01, 3.52)	0.12 (0.01, 1.86)	0.123
Non-central obesity	Ref	1.53 (0.90, 2.61)	1.58 (0.99, 2.53)	1.62 (1.03, 2.55) *	0.041 *
Central obesity	Ref	1.09 (0.62, 1.92)	1.17 (0.70, 1.96)	1.98 (1.18, 3.34) *	0.238
Cu (μg/L)	≤811.13	811.14–898.60	898.61–1005.97	≥1005.98	
Hyperuricemia/total	99/343	99/343	113/343	122/343	
BMI < 28 kg/m^2^	Ref	0.84 (0.57, 1.24)	1.09 (0.75, 1.60)	1.26 (0.85, 1.86)	0.121
BMI ≥ 28 kg/m^2^	Ref	0.57 (0.16, 1.99)	0.80 (0.20, 3.20)	1.76 (0.42, 7.43)	0.431
WHt R < 0.6	Ref	0.85 (0.59, 1.24)	1.17 (0.81, 1.70)	1.25 (0.85, 1.82)	0.105
WHt R ≥ 0.6	Ref	0.06 (0.01, 3.40)	0.06 (0.01, 4.84)	8.03 (0.12, 517.39)	0.156
Non-central obesity	Ref	1.05 (0.63, 1.76)	1.27 (0.77, 2.11)	1.94 (1.17, 3.23) *	0.007 *
Central obesity	Ref	0.59 (0.34, 1.02)	0.91 (0.52, 1.60)	0.78 (0.44, 1.38)	0.796
Zn (μg/L)	≤5141.11	5141.12–5906.57	5906.58–6781.64	≥6781.65	
Hyperuricemia/total	91/343	113/343	103/343	126/343	
BMI < 28 kg/m^2^	Ref	1.47 (1.00, 2.15) *	1.07 (0.73, 1.59)	1.28 (0.86, 1.90)	0.536
BMI ≥ 28 kg/m^2^	Ref	1.07 (0.28, 4.04)	1.39 (0.34, 5.73)	4.44 (0.99, 19.89)	0.056
WHt R < 0.6	Ref	1.45 (1.00, 2.11) *	1.13 (0.77, 1.65)	1.37 (0.93, 2.01)	0.285
WHt R ≥ 0.6	Ref	2.09 (0.10, 46.04)	2.55 (0.13, 50.67)	51.42 (0.53, 5013.22)	0.187
Non-central obesity	Ref	1.27 (0.77, 2.09)	1.04 (0.62, 1.73)	1.10 (0.65, 1.86)	0.955
Central obesity	Ref	1.59 (0.91, 2.75)	1.09 (0.62, 1.92)	1.74 (1.00, 3.02)	0.138
As (μg/L)	≤2.74	2.75–4.60	4.61–7.31	≥7.31	
Hyperuricemia/total	109/344	88/343	111/342	125/343	
BMI < 28 kg/m^2^	Ref	0.73 (0.50, 1.08)	1.03 (0.71, 1.50)	1.33 (0.89, 1.98)	0.090
BMI ≥ 28 kg/m^2^	Ref	0.30 (0.06, 1.48)	0.29 (0.07, 1.24)	0.30 (0.06, 1.49)	0.152
WHt R < 0.6	Ref	0.70 (0.48, 1.02)	0.94 (0.66, 1.36)	1.28 (0.87, 1.88)	0.133
WHt R ≥ 0.6	Ref	0.01 (0.01, 2.67)	1.06 (0.05, 21.33)	0.69 (0.04, 11.77)	0.911
Non-central obesity	Ref	0.96 (0.59, 1.56)	0.89 (0.54, 1.46)	1.07 (0.63, 1.82)	0.914
Central obesity	Ref	0.46 (0.26, 0.83) *	1.13 (0.65, 1.95)	1.37 (0.77, 2.42)	0.091
Female					
V (μg/L)	≤0.53	0.54–0.98	0.99–1.55	≥1.56	
Hyperuricemia/total	97/408	63/407	79/408	77/407	
BMI < 28 kg/m^2^	Ref	0.58 (0.39, 0.87) *	0.73 (0.47, 1.12)	0.67 (0.45, 1.00) *	0.121
BMI ≥ 28 kg/m^2^	Ref	0.39 (0.13, 1.13)	0.38 (0.12, 1.15)	0.69 (0.24, 1.93)	0.479
WHt R < 0.6	Ref	0.55 (0.37, 0.83) *	0.68 (0.44, 1.05)	0.73 (0.49, 1.09)	0.273
WHt R ≥ 0.6	Ref	0.64 (0.23, 1.77)	0.68 (0.24, 1.90)	0.42 (0.11, 1.53)	0.321
Non-central obesity	Ref	0.39 (0.21, 0.71) *	0.56 (0.30, 1.05)	0.74 (0.43, 1.29)	0.657
Central obesity	Ref	0.69 (0.42, 1.13)	0.75 (0.44, 1.26)	0.63 (0.38, 1.04)	0.103

Model is adjusted for age, education level, residence area, smoking status, alcohol use, physical activity, sitting time, diabetes mellitus, hypertension, TC, TGs, and metals in the multiple-metal model. ^a^ *p*-trend values across quartiles of metals is obtained by including the median of each quartile (natural log-transformed metals’ concentrations) as a continuous variable in the model. * *p* < 0.05.

**Table 5 nutrients-15-00552-t005:** Odds ratios and 95% confidence intervals for hyperuricemia according to the combined categories of whole blood metal concentrations and BMI in males.

	n (Hyperuricemia/Non-Hyperuricemia)	OR (95% CI)	OR- Int ^a^	RERI ^b^ (95% CI)	AP ^b^ (95% CI)	S ^b^ (95% CI)
Zn-BMI			2.28 (1.02, 5.06) *	1.67 (0.02, 3.33) *	0.56 (0.23, 0.88) *	6.02 (0.52, 69.37)
Low Zn + BMI < 28	178/446	Ref				
Low Zn + BMI ≥ 28	26/36	1.37 (0.77, 2.43)				
High Zn + BMI < 28	188/432	0.97 (0.74, 1.26)				
High Zn + BMI ≥ 28	41/25	3.01 (1.70, 5.31) *				
Cr-BMI			1.62 (0.73, 3.6)	1.58(−0.36,3.53)	0.44(0.08,0.81) *	2.62 (0.82, 8.4)
Low Cr + BMI < 28	160/467	Ref				
Low Cr + BMI ≥ 28	24/35	1.56 (0.87, 2.80)				
High Cr + BMI < 28	206/411	1.41 (1.09, 1.82) *				
High Cr + BMI ≥ 28	43/26	3.68 (2.12, 6.40) *				

Model is adjusted for age, education level, residence area, smoking status, alcohol use, physical activity, sitting time, diabetes mellitus, hypertension, TC, TGs, and metals in the multiple-metal model. ^a^ OR- int is assessed on the multiplicative scale by including cross-product terms in the model. ^b^ RERI, AP, and S are assessed on the additive scale. * *p* < 0.05.

## Data Availability

Not applicable.

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
