# Peer review of "Joint Effect of Multiple Metals on Hyperuricemia and Their Interaction with Obesity: A Community-Based Cross-Sectional Study in China"

_nutrients, 2023, doi:10.3390/nu15030552_

Round 1

Reviewer 1 Report

Thank you for your contribution.

I feel your research paper is very precise adn in-depth analysis of variosu metals on hyperuricemian and obesity.

I wonder Zn, Cr, Mn, etc are considered as elematary minerals not metals, which drive into misunderstanding as heavy metals. 

As this study is cross-sectional and uric acid is usually originated from food, this study can get readers have misunderstanding that several minerals can affect on hyperuricemia not by food containing uric acid.

None the less, this study is good to read for the understanding between the relation of some minerals and hyperuricemia, obesity by some inflammation and etc. 

Reviewer 2 Report

This is very interesting study that would be useful to provide further information on the status of multiple metals and their interaction of exposure on subjects with hyperuricemia and the effect on obesity. The manuscript is well written, and the findings are discussed and supported with appropriate figures and tables. There are minor comments as follows:

a)      Line 52, abbreviations are required for uric acid (UA) and in line 471 for reactive oxygen species (ROS).

b)      Line 120. The authors are commended for using reference material (SRM) (Seronorm) for quality control samples for method validation, which are important in assessing for accuracy and reliability of analytical data. However, the authors should consider to include the results of SRM in the table (e.g. in Table S1).

c)      Line 516. The authors are correctly mentioned that the study did not provide information on dietary exposure. Dietary exposure can contribute significantly to metal status in a population. These subjects could also be exposed to metals from drinking water and air pollution. Are the 11 districts selected for the study free from industrial or air pollution? There is a need of brief discussion on environmental exposure of metals in the study.
